# Economic Evaluation of Individualized Nutritional Support for Hospitalized Patients with Chronic Heart Failure

**DOI:** 10.3390/nu14091703

**Published:** 2022-04-20

**Authors:** Philipp Schuetz, Suela Sulo, Stefan Walzer, Sebastian Krenberger, Zeno Stagna, Filomena Gomes, Beat Mueller, Cory Brunton

**Affiliations:** 1Medical University Department, Kantonsspital Aarau, 5001 Aarau, Switzerland; happy.mueller@unibas.ch; 2Medical Faculty, University of Basel, 4001 Basel, Switzerland; 3Abbott Nutrition, Chicago, IL 60045, USA; suela.sulo@abbott.com (S.S.); cory.brunton@abbott.com (C.B.); 4MArS Market Access & Pricing Strategy GmbH, 79576 Weil am Rhein, Germany; stefan.walzer@marketaccess-pricingstrategy.de (S.W.); sebastian.krenberger@marketaccess-pricingstrategy.de (S.K.); 5Health Care Management, State University Baden-Wuerttemberg, 70174 Loerrach, Germany; 6Social Work & Health Care, University of Applied Sciences Ravensburg-Weingarten, 88250 Weingarten, Germany; 7Division of Diabetes, Endocrinology, Nutritional Medicine and Metabolism, Inselspital, Bern University Hospital, University of Bern, 4001 Bern, Switzerland; zeno.stanga@insel.ch; 8NOVA Medical School, Universidade NOVA de Lisboa, 1169-056 Lisboa, Portugal; filomenisabel@hotmail.com

**Keywords:** economic analysis, chronic heart failure, nutritional support, clinical outcomes, cost savings

## Abstract

Background Malnutrition is a highly prevalent risk factor in hospitalized patients with chronic heart failure (CHF). A recent randomized trial found lower mortality and improved health outcomes when CHF patients with nutritional risk received individualized nutritional treatment. Objective To estimate the cost-effectiveness of individualized nutritional support in hospitalized patients with CHF. Methods This analysis used data from CHF patients at risk of malnutrition (N = 645) who were part of the Effect of Early Nutritional Therapy on Frailty, Functional Outcomes and Recovery of Undernourished Medical Inpatients Trial (EFFORT). Study patients with CHF were randomized into (i) an intervention group (individualized nutritional support to reach energy, protein, and micronutrient goals) or (ii) a control group (receiving standard hospital food). We used a Markov model with daily cycles (over a 6-month interval) to estimate hospital costs and health outcomes in the comparator groups, thus modeling cost-effectiveness ratios of nutritional interventions. Results With nutritional support, the modeled total additional cost over the 6-month interval was 15,159 Swiss Francs (SF). With an additional 5.77 life days, the overall incremental cost-effectiveness ratio for nutritional support vs. no nutritional support was 2625 SF per life day gained. In terms of complications, patients receiving nutritional support had a cost savings of 6214 SF and an additional 4.11 life days without complications, yielding an incremental cost-effectiveness ratio for avoided complications of 1513 SF per life day gained. Conclusions On the basis of a Markov model, this economic analysis found that in-hospital nutritional support for CHF patients increased life expectancy at an acceptable incremental cost-effectiveness ratio.

## 1. Highlights

We previously reported a reduced risk for mortality and major cardiovascular events when older hospitalized patients with chronic heart failure and malnutrition received individualized nutritional interventions compared with similar patients who consumed only a usual hospital diet. In this study, we developed a Markov model of healthcare–state transitions and costs to identify the cost-savings and incremental cost-effectiveness ratios (ICER) of nutritional intervention. With an additional 5.77 life days, the overall ICER for nutritional support vs. no nutritional support was 2625 Swiss francs per life day gained.

## 2. Introduction

Chronic heart failure (CHF) has high clinical and economic costs worldwide given adverse health outcomes and increased healthcare resource utilization. Globally, HF cases exceed 60 million and account for nearly 10 million life-years lost to disability, with yearly costs estimated at nearly USD 350 billion [1,2]. The annual medical cost for a person with HF was estimated at more than USD 24,000 in the United States, although costs vary widely among individuals and are highest among those who are oldest and have co-morbidities [3]. Since HF imposes the greatest burden on older adults [1], the incidence is increasing as the population grows and ages [4].

Poor nutritional status is common among older people with HF because of multiple negative prognostic factors, such as decreased appetite and weight loss [5], impaired intestinal function [6], the presence of other comorbidities, and catabolic metabolism due to HF-related inflammation [7,8]. Malnutrition with consequent loss of muscle mass and physical functionality has been associated with increased morbidity, poorer quality of life, and worsening of CHF [9]. Nutritional strategies have long been recommended as part of treatment for CHF, but clinical studies often focus on restricting sodium intake and following specific dietary patterns for long-term cardiac health benefits, e.g., the Mediterranean and DASH diets [10,11].

Currently, many HF patients urgently need supportive nutrition care to address nutritional shortfalls and subsequent adverse consequences. Studies have reported improved health outcomes when patients with poor nutritional status receive nutritional interventions. In fact, quality improvement programs can be used across the continuum of care to enhance outcomes for people who have evidence of poor nutritional status in home-care settings, in residential nursing care [12], and during hospital admission [13,14,15,16,17]. An early review by Tappendan et al. found that hospital care with a focus on nutrition can reduce complication rates, length of hospital stays, readmission rates, and mortality [17]. Further, the results of a systematic review and meta-analysis of studies on hospitalized patients with malnutrition showed that nutritional interventions can significantly improve nutritional intake and reduce the risk of mortality [18]. Beyond health benefits, individualized nutritional support during and after hospitalization is also recognized as cost-saving because it spares healthcare resource utilization due to excess hospital lengths of stay, readmissions, and need for intensive care unit (ICU) admission [19,20,21,22]. In fact, the added cost of providing nutritional support is considered low, especially relative to the resultant lowered costs of hospitalization and medical treatments [20].

We previously reported results of beneficial health outcomes of nutritional intervention for at-risk patients in Swiss hospitals—a study known as Effect of Early Nutritional Therapy on Frailty, Functional Outcomes and Recovery of Undernourished Medical Inpatients Trial (EFFORT) [23]. In this study of more than 2000 medical inpatients, we found that nutritional interventions helped poorly nourished participants meet calorie and protein goals better than usual hospital food, significantly enhancing survival. When we focused the analysis on a subpopulation of EFFORT patients with CHF, we similarly found better health outcomes for the patients who were given supportive, individualized nutritional care [24]. Specifically, CHF patients at high nutritional risk had significantly reduced risk for mortality and major cardiovascular events when they received individualized nutritional interventions rather than standard hospital food [24]. In our current economic analysis of results from these vulnerable CHF patients in EFFORT, we applied a Markov model of health outcomes to predict how nutritional support would affect costs of healthcare utilization.

## 3. Methods

### 3.1. Study Design

This study was a secondary economic analysis of CHF patients who were part of EFFORT—a prospective, noncommercial, multicenter, randomized controlled trial. EFFORT was registered at ClinicalTrials.gov at https://clinicaltrials.gov/ct2/show/NCT02517476 (accessed on 7 August 2015) and conducted in eight Swiss hospitals. The overall objective of the original trial was to compare medical outcomes for patients at risk of malnutrition who were randomized to (i) an intervention group (individualized nutritional support to reach energy, protein, and micronutrient goals) or (ii) a control group (receiving usual hospital food).

Individualized nutritional support included screening patients for malnutrition risk on admission; dietitian-conducted nutritional assessment for patients identified to be at risk for malnutrition; individualized nutritional care plans developed by a dietitian; and implementation of the care plan with monitoring of health outcomes during hospitalization and follow-up post-discharge [23,25].

The rationale for the initial trial, design details, and eligibility features were previously reported [25], and the primary results of the full study were recently published [23,26], as were health outcomes in the CHF patient population [24]. The present study is based on CHF inpatients only, and it represents an analysis of healthcare costs and health outcomes in EFFORT’s two comparator groups—patients who were randomized to receive individualized nutritional support (intervention group) and those who received usual hospital food (control group) [24]. EFFORT included a total of 645 patients with CHF, with 234 (36%) acutely decompensated and 411 (64%) with chronic stable HF [24].

### 3.2. Health Economic Terms Used

Here, we provide definitions of key health economic terms (Appendix A, Table A1) used in our report [20,27,28].

### 3.3. Description of Markov Simulation Model

We developed a Markov simulation model with daily cycles to analyze the economic impact of nutritional support in malnourished inpatients with CHF; the model reflected the perspective of Swiss health insurers. A modeling timeframe of six months (180 days) with five designated health states was based on findings in a recent systematic review and meta-analysis report [18]. In the present analysis, we assumed that all patients began in a stable health state—hospitalization with HF and evidence of malnutrition risk on admission (Figure 1). During hospitalization, patients could develop complications, such as myocardial infarction or arrhythmia. This complication state was modeled as an autonomous state because the probability of death is higher than for patients not experiencing in-hospital complications. Worsening CHF and complications might require transfer to the ICU. Other modeled states included discharge from the hospital and readmission for a non-elective reason. Notably, patients had different costs for care and risks of death in each state. Transition probabilities between health states were based on the outcome results for CHF patients in our full EFFORT clinical study [24]. Transition values are compiled in Table A2 of Appendix A). Raw data were taken from the original EFFORT study for the CHF population and then put manually into the simulation model via Excel.

### 3.4. Patient Population

For the initial trial, we screened medical patients upon hospital admission for risk of malnutrition using the Nutritional Risk Screening (NRS) 2002 [29]. We included adult patients with a total NRS score ≥ 3 points, an expected length of stay (LOS) > 4 days, and written informed consent. We excluded patients who were treated in the intensive care or surgical units, were unable to have oral intake, or were receiving long-term nutritional support on admission; patients with terminal illnesses, gastric bypass surgery, anorexia nervosa, acute pancreatitis, acute liver failure, cystic fibrosis, stem cell transplantation; and patients previously included in the trial. All patients eligible for this secondary analysis had a documented diagnosis of CHF on hospital admission, which was confirmed and validated by a complete chart review after hospital discharge. In line with the European Society of Cardiology (ESC) guidelines [29], we stratified CHF patients, according to their ejection fraction, into three groups: (1) reduced ejection fraction (HFrEF; rEF < 40%), (2) mid-range ejection fraction (HFmrEF; mr EF 40–49%), and (3) preserved ejection fraction (HFpEF; pEF ≥ 50%).

Table A3 of Appendix A gives an overview of the main results from the initial report [24].

### 3.5. Costs and Utilities

Utility values (cost of gained effectiveness of nutritional support) were derived from a study by Schuetz et al., assuming the utility value for preventing a major cardiovascular event (MACE) was a reasonable proxy for developing a major complication (adverse event) during hospitalization [24,26]. Costs for the different health states were assumed as follows: (1) costs for nutritional inpatient support were based on the publication by Schuetz et al. 2020 [26], assuming a standard deviation of 20% of the input value, for both in- and outpatient nutritional support; (2) costs for 20% of post-discharge patients to continue nutritional supplements were based on cost data from the largest Swiss online pharmacy [30]; (3) costs for a heterogeneous distribution of cardiovascular events were estimated on the basis of the Swiss Disease-related Group (DRG) costs for severe arrhythmia and cardiac arrest [31]; (4) ICU costs were based on the Swiss DRG costs for an intensive care complex treatment [31]; and (5) no costs were assigned for death (Table 1).

### 3.6. Base-Case and Cost-Effectiveness Analyses

The primary outcomes in our model were *cost-by-health-state* and *total cost*. We calculated days in each health state and calculated utility values as the difference between the total costs of individualized nutritional support compared with no support. Because real-life findings were modeled, we did not apply any discount rates.

### 3.7. Sensitivity Analyses

Since costs of nutritional supplements may vary in different health states and care sites, we performed a sensitivity analysis to determine whether cost savings would be maintained when the costs of nutritional supplements were 5 SF per day (lower bound), 100 SF per day (medium bound), and 1000 SF per day (upper bound).

Further, we ran sensitivity analyses (1) assuming 50% of discharged patients would continue oral nutritional support in the outpatient setting (5 SF per day, corresponding to one oral supplement per day) and (2) assuming 100% of discharged patients would continue nutritional support in the outpatient setting (5 SF per day). We also analyzed the costs per life-year. Therefore, we extrapolated the data from 180 days to 365 days. Finally, we investigated which costs for nutritional support would still be cost-effective at a threshold of 100,000 SF per life-year.

We followed the international modeling guidelines of the ISPOR SMDM Modeling Good Research Practices Task Force [33,34] and the reporting recommendations of the Consolidated Health Economic Evaluation Reporting Standards (CHEERS) statement [35].

## 4. Results

### 4.1. Patient Outcomes

In the original analysis of the EFFORT trial, 645 patients had CHF (321 patients allocated to the intervention group and 324 patients allocated to the control group). Compared with patients in the control group, the 180-day mortality rate for patients who received nutritional support was significantly lower (85 of 321 (26.5%) vs. 102 of 324 (31.5%)) with an adjusted hazard ratio of 0.74 (95% CI: 0.55 to 0.996; *p* = 0.047) [24].

### 4.2. Base-Case Analyses of Cost-Effectiveness

A base-case analysis summarizes our cost results (Table 2). Here, the term ‘Life days’ represents the number of patient days in each health state. Utility results are shown as quality-adjusted life days (QALDs), which were calculated in the model. Finally, the calculated costs for each health state are shown. The per-patient costs for in-hospital nutritional support were estimated at 679 SF (EUR 651) per patient across the patient’s hospital length of stay. In terms of costs over the 6-month timeframe of the study model, hospital care averaged 229,036 SF (EUR 219,427) per patient in the intervention group versus 213,878 SF (EUR 204,905) in the control group. These totals included costs for days in the normal ward, days in the ICU, and added costs due to complications. Ongoing nutritional support in the outpatient setting amounted to 19 SF (EUR 18) in total since 20% of the patients continued oral nutrition supplements after discharge from the hospital. Sensitivity analysis within a range of 5 SF to 1000 SF per day for nutritional supplements did not overcome the cost benefit for nutritional support at a threshold of 100,000 SF per life-year.

Incremental differences in cost, life days, and the incremental cost-effectiveness ratio (ICER) were determined (Table 3). When using nutritional support, the total cost difference over the 6-month modeling interval was 15,159 SF (EUR 14,523), which was mainly driven by increased days in a normal ward (20,798 SF) and by cost savings due to avoided complications (6214 SF). In terms of complications, patients receiving nutritional support had 4.11 more life days without complications. Given the cost savings of 6214 SF (EUR 5953) and the additional 4.11 life days, the ICER per avoided complication was 1513 SF (EUR 1450). The overall ICER for nutritional support vs. no nutritional support was 2625 SF (EUR 2515) per life day saved.

### 4.3. Sensitivity Analyses

Even when varying input values for sensitivity analyses, findings were consistent with the original analysis (Appendix A, Table A4). When adjusting the proportion of patients continuing nutritional support after being discharged from the hospital, no relevant increases in nutrition costs could be observed. With 50% of patients receiving outpatient nutritional support, 47 SF (EUR 45) would have to be invested for 180 days, and 134 SF (EUR 128) would have to be invested for one year. With 100% of patients, those costs would amount to 94 SF (EUR 90) per 180 days and 269 SF (EUR 258) per year. We also analyzed different cost input values for nutritional support and the maximum cost input to stay under a threshold of 100,000 SF per life-year. The maximum cost input would be 6755 SF (EUR 6472) if 100% of patients continued nutritional support in the outpatient setting; 7497 SF (EUR 7182) if 50% of patients continued nutritional support as outpatients; and 8027 SF (EUR 7690) if only 20% of patients continued nutritional support as outpatients.

## 5. Discussion

In our prior study of hospitalized CHF patients with malnutrition (or risk of malnutrition) receiving nutritional support, we reported a significantly reduced risk for mortality and major cardiovascular events compared with CHF patients who consumed the usual hospital diet [24]. Importantly, the results of our current modeling study showed that the added cost of providing nutritional support is relatively low, especially when considering the associated reduction in risk for complications and their excess costs (extended hospitalization time and more medical treatments). Altogether, the results from our present Markov healthcare cost modeling for hospitalized CHF patients showed that nutritional care (i.e., in-hospital nutritional support continued post-discharge as needed) is a cost-effective intervention. This finding underscores the benefits of routine and robust nutritional intervention for all patients hospitalized with CHF, i.e., screening patients for malnutrition or its risk when they are admitted to the hospital, then providing nutritional support according to a dietitian-recommended, individualized plan. While the focus of our study and others was on healthcare utilization and cost, we note that such cost savings occur in the context of improved patient outcomes, especially longer survival [23].

Nutrition interventions for hospitalized patients have been established as cost-effective strategies that also yield benefits in terms of better patient outcomes, especially for older adults [36,37]. In terms of health economics, value is determined as outcomes relative to costs; in the value equation, the numerator is the outcome, while the denominator is the cost. Depending on the stakeholder’s perspective, high value may be viewed as reduced patient morbidity and mortality, cost containment, or profitability [38]. All stakeholders recognize the value of better patient health outcomes.

Rising healthcare expenditures necessitate the adoption of evidence-based strategies for cost containment, especially for hospital care. The strategy of improving patients’ nutritional status to improve health and cost outcomes is well-known and gaining ever-growing supportive evidence. In a recent systematic review, Galekop et al. identified 53 studies that analyzed the cost-effectiveness of personalized nutrition in patient care [39]. Nearly half of the analyses (49%) concluded that nutritional intervention was cost-effective, and 75% of the incremental cost–utility ratios were cost-effective given a willingness-to-pay threshold of USD 50,000 per quality-adjusted life-year [39]. Other researchers performed a specific value analysis on the use of nutritional support therapy to lower the risk of hospital-acquired infections (HAIs), which are life-threatening and expensive to treat [40]. On the basis of decreased HAIs and the shortened length of hospital stay among patients who were critically ill or undergoing major surgery, these researchers reported that nutritional support therapy has the potential to save the United States (US) Centers for Medicare and Medicaid Services approximately USD 104 million annually [40]. A broader Medicare Claims modeling study, the Value Project of the American Society for Enteral and Parenteral Nutrition (ASPEN), projected annual cost savings from nutritional support therapy in five selected therapeutic areas—sepsis, gastrointestinal cancer, hospital-acquired infections, surgical complications, and pancreatitis [41]. The total cost savings was estimated at USD 580 million per year [41]. Another research team conducted an economic evaluation alongside a multicenter randomized controlled clinical trial (the NOURISH Study); the study population was malnourished older patients in US hospitals [42]. Across a 90-day time horizon, nutrition therapy yielded health improvements at a cost of no more than USD 34,000 (EUR 29,800) per quality-adjusted life-year. When extending the time horizon to a patients’ entire lifetime, the intervention cost only USD 524 (EUR 460) per life-year saved [42].

However, disease-associated malnutrition often remains undiagnosed and untreated. While medical nutritional support requires multidisciplinary awareness and care, Meehan and colleagues noted that hospital nurses are ideally positioned to play critical roles in nutrition—screening for malnutrition on patient admission to the hospital, monitoring for and addressing conditions that impede nutrition intake, and ensuring that prescribed nutritional interventions are delivered and administered or consumed [14]. Such nursing support in multidisciplinary nutrition care can contribute to better patient outcomes at lower costs [14].

Our economic analysis model has limitations inherent to most modeling analyses. Costs and cost savings were calculated from the perspective of the 27 hospitals included in the Gomes et al. review and meta-analysis [18]; the results may thus not be fully generalizable to other hospitals. Demographics and different levels of need for care could have influenced treatment outcomes and related costs. Populations are becoming increasingly older, and elderly patients are perceived to need more care support. However, only total costs would be influenced by this need for care. Incremental costs would remain the same, as these patients have a need for additional care independent of the nutritional intervention. In addition, concomitant and other diseases could cause additional costs and influence the outcome of CHF treatment. Further, our cost data and reported savings are calculated from the perspective of Swiss hospital payers and their reimbursement system; this model may not be generalizable to other hospitals or to the outpatient setting. The ICER of 100,000 SF used in our sensitivity analysis is hypothetical because in Switzerland, no cost-effectiveness threshold is applied in reimbursement decisions. Finally, our model uses direct costs as the main drivers of economic decision-making from the perspective of hospital administrators and payers; future models could tackle savings in cost terms important to the patients, such as faster recovery with less disability and lower loss of work productivity.

## 6. Conclusions

This Markov-modeled economic analysis showed that in-hospital nutritional support for chronic HF patients with malnutrition was a cost-effective strategy to improve health outcomes. Compared with other more invasive procedures, nutritional support is easy to implement in hospitals and other care settings and can help protect patients from adverse events that require cost-intensive interventions, such as 21,750 SF (EUR 20,838) for a coronary bypass or 27,818 SF (EUR 26,651) for cardiac defibrillator implants [43].

### 6.1. Clinical Perspective

Given the high proportion of older people with HF and at risk of malnutrition [9,44], we anticipate that patient-specific nutritional interventions can lead to substantial reductions in healthcare costs in addition to well-recognized health and mortality benefits. The evaluation of other patient-centered outcomes, such as quality of life, should also be explored in future studies.

### 6.2. Translational Outlook

The significant reduction in hospital complications and the associated costs in the subgroup of HF patients with established malnutrition may be particularly relevant for policymakers. We anticipate that such findings will be confirmed and extended by randomized controlled trials that specifically enroll hospitalized patients with CHF.

## Figures and Tables

**Figure 1 nutrients-14-01703-f001:**
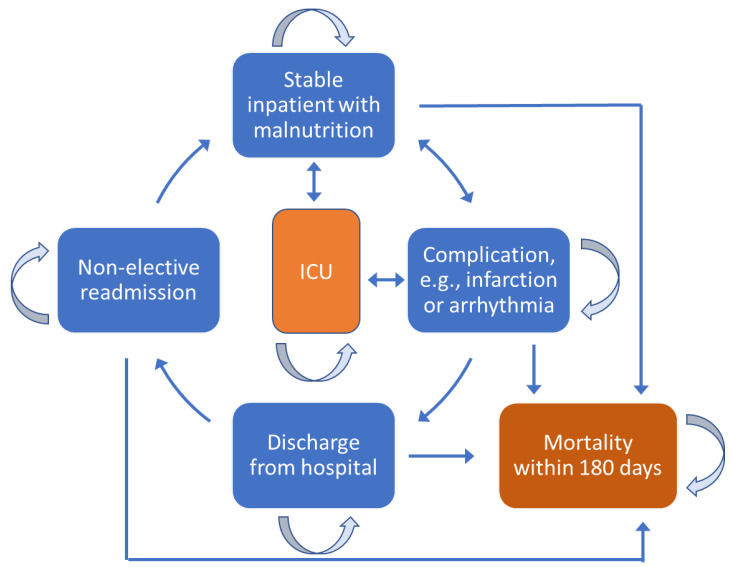
Health states of the Markov model. Light blue arrows represent patients staying within the given health state, while bright blue arrows represent transitions between states. Abbreviation: ICU, intensive care unit.

**Table 1 nutrients-14-01703-t001:** Cost input values for the health economic model with monetary costs expressed in Swiss francs (SF).

Cost Item	Cost Input, Swiss Francs (SF)	For Probabilistic Analysis		Reference
		Distribution	SD, (SF)	
**Nutritional support inpatient**	5	Gamma	1	ZRMB [30]
**Nutritional support outpatient**	5	Gamma	1	ZRMB [30]
**Cost per day in normal ward**	1650	Gamma	1485	BFS 2020 [32]
**Cost per day in ICU**	4654	Gamma	3900	DRG [31]
**Average cost per complication (per day)**	1513	Gamma	1477	DRG [31]

ICU: intensive care unit; SD: standard deviation; SF: Swiss francs. Costs were rounded to the nearest full unit. 1 SFCHF = 0.95 EUR.

**Table 2 nutrients-14-01703-t002:** Costs and cost differences by nutrition group over 180 days for HF patients in the EFFORT trial.

	Life Days		Utilities		Cost (Swiss Francs, CHF)	
Cost Item	Individualized Nutritional Support	No Nutritional Support	Individualized Nutritional Support	No Nutritional Support	Individualized Nutritional Support	No Nutritional Support
**Nutrition (support)**					679	--
**Days in normal ward**	123.84	111.24	0.25	0.23	204,342	183,544
**Days in ICU**	1.88	1.90	0.00	0.00	8733	8857
**Complications**	10.09	14.20	0.02	0.03	15,263	21,477
**Post-hospital discharge life days**	18.77	21.47	0.04	0.04	19	0
**Total**	154.58	148.81	0.31	0.30	229,036	213,878
**Difference**	**5.77**		**0.02**		**15,159 SF**	

ICU: intensive care unit; SF: Swiss francs. Costs were rounded to the nearest whole unit. All other data were rounded to two decimal places. 1 SF = EUR 0.95.

**Table 3 nutrients-14-01703-t003:** Results for incremental differences from base-case analysis of HF patients in EFFORT.

	Incremental Changes for Nutritional Support vs. No Nutritional Support		
Cost Item	Cost, Swiss Francs (SF)	Life Days	ICER LD, SF
**Day in normal ward**	20,798	12.60	1650
**Day in ICU**	−123	−0.03	4109
**Complication (AE)**	−6214	−4.11	1513
**Post-hospital stay, life days**	19	−2.70	−7
**Total**	15,159	5.77	2625

AE: adverse event; ICER LD: incremental cost-effectiveness ratio per life day; ICU: intensive care unit; costs were rounded to the nearest full unit, and all other data were rounded to two decimal places. 1 SF = EUR 0.95.

## Data Availability

The data presented in this study are available on request from the corresponding author. The data are not publicly available due to privacy and ethical reasons.

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
