# Peer review of "Economic Evaluation of Individualized Nutritional Support for Hospitalized Patients with Chronic Heart Failure"

_nutrients, 2022, doi:10.3390/nu14091703_

Round 1
Reviewer 1 Report
This is a retrospective study regarding the cost-effectiveness analysis of individualized nutritional support in hospitalized patients with HF suing the data of RCT (EFFORT study). As the already published findings of EFFORT study, nutritional support lowered 180-day mortality. The Authors calculated cost-effectiveness ratio for nutritional support vs. control was 2,625 SF per life day. The Authors concluded that nutritional support was effective with acceptable cost-effective ratio.
Major comments:
- Presented data were mostly cost-effective analysis. But cost-effectiveness is based on the beneficial/harmful effect of nutritional support on clinical outcome (events, complications, ADLs or length of hospital stay etc.). The Authors need to show these basal data related to cost-effectiveness analysis. First of all, how many patients were allocated to each group? And, for example, how many days did nutritional support group/no nutritional support group stay within each health state in the Markov model? How were those data extracted from the raw data? The Reviewer assumes that a part of these data are shown in Table 2a. But it is hard to interpret and the Authors need to show them so that the readers can easily understand.
- The Authors stated that 100000SF per life year was set as a threshold of cost-effectiveness. Please explain why ICER 2625 SF per life day was acceptable. How was this threshold defined?
- Table 2 is confusing. What does “Utilities (QALDs) mean? The Reviewer assumes that “Life days” might correctly mean “QALD”, rather.
- Introduction is redundant. Especially, most of the first paragraph can be omitted.
Minor comments:
Line 137. Myocardial or cerebral “infarction”?
Table 2a. The Reviewer cannot see the right side of the Table.
Reviewer 2 Report
The authors present an economic evaluation of individualized nutritional support for hospitalized patients with chronic heart failure. This is an extremely interesting study since it analysis an easy-to-implement strategy to improve outcomes. The manuscript is very well written and easy to read. I have some comments:
- According to the authors, EFFORT included 645 patients with CHF, with 234 (36%) acutely 124 decompensated and 411 (64%) chronic stable HF. Did the economic analysis yield different results in patients with acutely decompensated HF than those with CHF?
Round 2
Reviewer 1 Report
Thank you for the amendement suggested by the Reviewer. No further comments.